# Fatty Acids Present in Wheat Kernels Influence the Development of the Grain Weevil (*Sitophilus granarius* L.)

**DOI:** 10.3390/insects12090806

**Published:** 2021-09-08

**Authors:** Mariusz Nietupski, Emilia Ludwiczak, Robert Cabaj, Cezary Purwin, Bożena Kordan

**Affiliations:** 1Department of Entomology, Phytopathology and Molecular Diagnostics, University of Warmia and Mazury in Olsztyn, 10-719 Olsztyn, Poland; emilia.bujak@uwm.edu.pl (E.L.); robert.cabaj@uwm.edu.pl (R.C.); bozena.kordan@uwm.edu.pl (B.K.); 2Department of Animal Nutrition and Feed Science, University of Warmia and Mazury in Olsztyn, 10-719 Olsztyn, Poland; cezary.purwin@uwm.edu.pl

**Keywords:** stored product pest, natural resistance, granary weevil, fatty acids

## Abstract

**Simple Summary:**

The grain weevil (*Sitophilus granarius* L.) is a common storage pest species, whose foraging on stored cereal grain causes major losses worldwide every year. Traditional ways to control this species (non-chemical) are expensive or do not guarantee effective control against this pest insect. The application of chemical methods, on the other hand, is contradictory to consumers’ expectations regarding food safety, and cause a negative impact on the natural environment. An important step in the development of a safe and effective strategy for limiting the losses caused by the grain weevil is to search for mechanisms influencing the natural resistance of cereal grain and then to use this knowledge in the breeding of new cultivars. This experiment entailed assessing the content of fatty acids in grains of selected wheat cultivars. The content of these compounds and their composition in the wheat cultivars varied. In addition, the life cycle of *S. granarius* on the tested wheat grain was assessed. The results of the experiment suggested that the intensity of the development of *S. granarius* is significantly correlated with a higher content of saturated fatty acids and unsaturated ones in kernels. The substances that might stimulate the development of the grain weevil or reduce the number of offspring of this beetle were identified.

**Abstract:**

*Sitophilus granarius* (L.) is considered to be one of the major pests causing damage to cereal grain stored in silos and granaries. Using traditional methods (synthetic insecticides, mechanical, or physical methods) to control this pest is either ineffective or dangerous to people and nature. It is, therefore, necessary to develop new cultivars of cereals that will be distinguished by a high natural tolerance of the foraging by *S. granarius*. The aim of this study is expressed in the set research hypothesis, stating that the number of offspring of the grain weevil on stored wheat kernels can depend on the content of fatty acids in the kernels. Thus, the qualitative and quantitative composition of fatty acids was determined in kernels of 10 winter wheat cultivars, and the abundance of the beetle’s offspring generation of *S. granarius* that developed on the wheat grain, as well as the mass of produced dust and loss in the mass of wheat grain were determined. By applying statistical analyses (GLM, ANOVA, Pearson’s linear correlation coefficient, and analysis of redundancy), the presence and character of the dependence between the determined content of fatty acids in wheat grain and the factors describing the development of *S. granarius* were established. The research results indicate that fatty acids from the groups C 18:1 and C 20:1 probably play an important role as substances stimulating the increase in the number of the tested pest progeny. In contrast, fatty acids C 15:0, C 16:1, and C 18:3, which were determined in large amounts in the grain of wheat cultivars Speedway, KWS Livius, and Julius, can reduce the number of offspring of pest insect.

## 1. Introduction

Wheat (*Triticum* L.) is among the major cereal plants, which are sources of protein, a fundamental nutrient in the human diet [1,2,3]. Wheat is the staple food source for 40% of the global population and is therefore considered a strategic product in the food economy [4]. Reducing losses at all stages, cultivation, harvest, as well as storage, is an important element in the wheat production process that influences wheat yields. Despite the high content of dry matter in grain, which favors long-term storage, it is still difficult to store wheat well, especially in the case of insufficiently dried wheat grains (humidity above 8% promotes the growth of mold, bacteria, mites, and insects). The main factor that leads to qualitative and quantitative losses of cereal grain in granaries and silos is foraging by insects [5]. According to Mebarkia et al. [1], in some countries, up to 50% of harvested yields are lost due to pest insects (the high humidity of the grain and its inadequate storage allow pest access). The growing demand for food, including organic food and the increasing tolerance of insects to the major insecticides used, is the reason why research on preventive pest control methods gains importance [6,7,8].

Among numerous storage pests, the beetle from the Curculionidae family called the grain weevil (*Sitophilus granarius* L.) is the one that occurs more frequently, causing up to 5% damage to stored grain [9,10,11,12]. This pest is distributed throughout the temperate regions of the world, and in tropical climates, it occurs only in cool upland areas. The foraging by this insect on cereal grain decreases the grain mass, deteriorates its nutritive and commercial value, and reduces the germination capacity [13]. Traditional methods of controlling this species are based on chemical methods, but the application of broad-spectrum contact insecticides can have a negative impact on human health and the environment, in addition to which it increasingly often induces the emergence of tolerance of the pests to insecticides [14,15]. The development of the grain weevil is strongly influenced by the temperature, moisture, pest density, and type of food [16]. Under relatively stable abiotic conditions present in granaries and silos for storing grains, the chemical properties of grain can be a key factor in the natural tolerance of grain to the foraging by *S. granarius*. There are studies confirming that the extent of damage to stored wheat grain by *S. granarius* varies and is connected with cultivar-specific characteristics of wheat [1,17]. Factors that determine the vulnerability of certain cereal varieties to the foraging by storage pests are both the physical qualities of kernels (e.g., grain hardness, glassiness, the thickness of the seed cover) and biochemical ones [18,19,20,21,22]. Gowda et al. [23] conclude that the chemical properties of rice grains significantly affect the development of *Sitophilus oryzae* L. Nwosu [24] showed that the resistance of maize kernels to the foraging by *Sitophilus zeamais* Motsch. depended on the chemical composition of kernels. For *S. granarius*, starch is the primary source of energy, while protein is used as the main building material [19]. Lemic et al. [25] draw attention to the lack of detailed studies dealing with the nutritional preferences of the grain weevil when selecting grain based on a detailed description of the grain’s chemical composition. Lipid type substances contained in the outer layer of a kernel can play a key role in the process of searching for food [26], and subsequently, in the laying of eggs by female beetles [19,21,27,28]. Fatty acids (saturated fatty acids, monounsaturated fatty acids, and polyunsaturated fatty acids) are a group of substances that are associated with the chemical characteristics of wheat kernels, providing them with resistance to the infestation by *S. granarius* and *Rhyzopertha dominica* F. [29].

Lipids and fatty acids, however, are present in both the outer layer of a kernel and in its embryo, in the bran (especially in the aleurone layer), endosperm, and starch granules [30]. The content of these bioactive compounds in cereal grain varies and depends on several factors, especially the species and cultivar of a cereal, meteorological conditions, and cultivation technologies. The most important fatty acids of durum wheat (*Triticum durum* Desf.) are linoleic acid (C 18:2), palmitic acid (C 16:0), oleic acid (C 18:1), linolenic acid (C 18:3), stearic acid (C 18:0), and palmitoleic acid (C 16:1) [31]. Kordan et al. [17] maintain that C 20:1 is a fatty acid that strongly attracts the grain weevil. Uncovering the qualitative and quantitative composition of fatty acids in different wheat cultivars and connecting this knowledge with data on the grain weevil’s development can provide important information for the breeding of wheat cultivars tolerant to the foraging of *S. granarius*.

The research hypothesis set in this study is that the development of the grain weevil on stored wheat kernels can be affected by the fatty acids that are contained in these kernels. An analysis of this dependence and its character are entailed the following steps:determination of the qualitative and quantitative composition of fatty acids in kernels of ten winter wheat cultivars,assessment of the number of offspring of *S. granarius*, mass of produced dust, and loss of grain mass on the tested wheat cultivars,identification of the presence and character of a relationship (positive or negative) between the content of determined fatty acids in wheat grain and the parameters describing the development of *S. granarius* listed above.

## 2. Materials and Methods

### 2.1. Materials

The laboratory investigations were carried out at the Chair of Entomology, Phytopathology and Molecular Diagnostics of the University of Warmia and Mazury in Olsztyn, Poland. The research material consisted of kernels of 10 winter wheat cultivars (Bogatka, Muszelka, Ostroga, Askalon, Bamberka, Forum, Speedway, Platin, KWS Livius, Julius) obtained in 2014 from the Experimental Station for Evaluation of Cultivars in Radostowo (northern part of Poland). The tested grain was conditioned in a plant growth chamber (Sanyo MLR—350 H – Sanyo Electric Co., Ltd., Japan) for 7 days at the temperature optimal for the tested beetle species (27 °C, relative air humidity 75%). Afterwards, the analysed material was passed through a 1 mm mesh sieve (to sift and remove the dust) and 20 g samples were weighed. Each sample was placed in a PVC-U container size 8 cm in diameter and 3 cm in height. A ventilation hole of 2 cm diameter was drilled in the cap. The hole was covered with chiffon mesh to prevent the beetles from escaping the container.

### 2.2. Analysis of the Composition of Fatty Acids (FA)

Methyl esters of fatty acids were prepared according to a modified method originally developed by Peisker (methanol: chloroform: concentrated sulphuric acids in the ratio of 100:100:1 *v*/*v*) [32] for the determination of the composition of fatty acids. The separation and determination of fatty acids followed the gas chromatography method using a Varian CP–3800 gas chromatography (Varian Instruments, Palo Alto, CA, USA) coupled with a flame ionisation detector (FID), capillary column 50 m in length (inner diameter of 0.25 mm, thickness of film 0.25 μm), and a split injector (50:1 split ratio). The volume of a sample placed in the chromatograph was 1 μL. The injector temperature was 250 °C, and the column temperature was 200 °C. The carrier gas was helium (1.2 mL/min flow rate). Identification of fatty acids was achieved by comparing the time of retention of individual patterns of fatty acid methyl esters (Sigma_Aldrich, Saint Louis, Missouri, United States) and peaks in each sample. The relative content of fatty acids was expressed as a % of the total area of all fatty acids contained in the sample. Determinations of chemical parameters of kernels of the tested winter wheat cultivars were made at the Chair of Animal Nutrition and Feed Science, which belongs to the Faculty of Bioengineering at the University of Warmia and Mazury in Olsztyn. The analysed acids fall into three groups. One is composed of saturated fatty acids (SFA), another one consists of monounsaturated fatty acids (MUFA), and the last one encompasses polyunsaturated fatty acids (PUFA). This division was proposed—as explained by [33]—based on the number of double bonds between carbon atoms present in these compounds.

### 2.3. Bioassays

The entomological material used in this experiment originated from mass breeding of *S. granarius*, from which adult specimens were obtained. Conservation breeding of this species was carried out on the grain of the winter wheat cultivar Korweta. The adult pests used in this experiment originated from the mass breeding of *S. granarius*. It is a long-standing (maintained since the 1990s, annually renewed the gene pool by introducing new individuals from granaries located in north-eastern Poland) laboratory strain of *S. granarius*, which has been bred at the Department of Entomology, Phytopathology and Molecular Diagnostics, University of Warmia and Mazury in Olsztyn for many years. Entomological observations took place in strictly controlled conditions in a plant growth chamber SANYO MLR 35O-H at constant temperature (27 °C) and humidity (relative air humidity 75%) and in complete darkness [16].

Twenty adult specimens of the grain weevil (3–4 days old), in a 1:1 sex ratio, were placed on each of the prepared grain samples. The sex of beetles was identified by examining the proportions of the rostrum and the shape of the 5th and 6th abdominal sternite [34]. The experiment was performed with 10 replicates for each wheat cultivar. There was no control combination in the experiment because we investigated the effect of fatty acids within different cultivars of one type of grain—wheat. After 8 weeks, based on live and dead *S. granarius* specimens, the number of specimens in the offspring generation was counted. Both the produced dust and kernels were weighed on a laboratory balance WPS 220/C/2 (Radwag, Radom, Poland), thereby determining the loss of grain mass. Based on our previous experiments [21,35], we presumed that we would terminate this experiment 8 weeks after placing adult forms on wheat grain. According to Gołębiowska [16], most of the new generation adults of *S. granarius* emerge, on average, between 8 and 9 weeks since the oviposition.

### 2.4. Statistical Analysis

An evaluation of the character of the distribution of data concerning the number of offspring specimens, loss of mass of kernels, the mass of the dust generated by the beetles, and content of fatty acids in kernels of the tested wheat cultivars was based on W Shapiro–Wilk’s test. Differences between the means characterised by the unimodal distribution were assessed using a generalised linear model (GLM), taking into account Poisson’s distribution of data. Data that presented normal distribution were evaluated with one-factorial ANOVA. Groups of means of the parameters connected with the development of *S. granarius* that were not statistically different were labelled with the same letter index, i.e., a, b, c, …. (Tukey’s HSD test). Pearson’s linear correlation coefficient *r* was computed to determine the relationships between the content of fatty acids in kernels and the parameters describing the development of *S. granarius*. This coefficient expressed the strength of a correlation between two variables and is given as a numerical value with the interval [–1, 1]. The sign of coefficient *r* indicates the direction of a correlation, while its absolute value expresses the power of this relationship. The significance of this coefficient was also tested by calculating the test probability value *p*. Graphic presentation of the achieved assessment results employed the use of ordinance techniques [36]. The RDA (redundancy analysis) was applied because the distribution of the analysed data was linear (SD = 0.1). All statistical calculations and their graphic interpretation were done using the following software: Statistica 13.1 (Dell Inc. Tulusa, OK, USA) and Canoco 4.51 (Biometris – Plant Research International, Wageningen, The Netherlands).

## 3. Results

### 3.1. Parameters of the Development of S. granarius

Parameters describing the intensity of the development of *S. granarius* on the tested wheat grain were evaluated by assessing the abundance of the pest’s offspring generation (Wald’s Stat. = 4227.40, *p* = 0.00), the mass of the dust produced (W = 31.82, *p* = 0.00), and loss of the mass of kernels (W = 108.53, *p* = 0.00). The means followed unimodal distribution, which is why the generalised linear model (GLM) was applied for the assessment of the significance of differences. It showed that differences between the mean values of the analysed parameters were statistically significant (Table 1).

The highest number of *S. granarius* offspring was found on the grain of the three cultivars: Askalon (375.2 specimens on average), Bamberka (369.4), and Ostroga (347.3) (Figure 1a). The cultivars on whose kernels the grain weevil developed most poorly were KWS Livius, Bogatka, Speedway, Platin, and Julius, where the number of offspring beetles varied in the range of 159.7–125.6 of individuals. The mass of dust produced by *S. granarius* beetles followed a similar pattern (Figure 1b). The cultivars on which this pest produced most dust were Askalon (2.4 g on average), Bamberka (2.2 g), and Ostroga (2.2 g). Significantly less dust was determined in the treatments with kernels of the cultivars Forum and Muszelka. The least dust (0.78–0.61 g) was found in the tests with the cultivars KWS Livius, Speedway, Platin, Julius, and Bogatka. Foraging beetles and larvae of the grain weevil caused the highest loss of grain mass on the cultivars Askalon, Bamberka, and Ostroga (12.33–11.78 g on average) (Figure 1c).

Significantly lower values of this parameter were determined for the kernels of the cultivars Muszelka (10.77 g on average) and Forum (10.06 g on average). Finally, the smallest loss of the mass of kernels (5.60–4.95 g) appeared in the variants with the cultivars Speedway, KWS Livius, Bogatka, Platin, and Julius.

### 3.2. Content of Fatty Acids in Wheat Kernels

The wheat grain on which *S. granarius* developed was submitted to an analysis of the content of fatty acids (FA). In total, the presence of 14 substances was detected. They differed in the length of the carbon chain, and their content in the analysed wheat cultivars was statistically significantly different (Table 2).

The most numerous fatty acids were C 18:2, C 16:0, and C 18:1 (Table 3). Their high content was determined in the grain of cv. Bogatka (C 18:2, C 18:1), Muszelka (C 16:0), Ostroga (C 16:0, C 18:1), and Bamberka (C 18:1). The fatty acids that appeared in the smallest amounts in wheat kernels belonged to the groups C 12:0, C 14:0, and C 15:0. Their highest concentrations were identified in kernels of cv. Bamberka (C 12:0), Bogatka (C 14:0), and Muszelka (C 15:0). They were the least abundant in kernels of cv. Julius. The isolated fatty acids represented saturated (SFA) and unsaturated fatty acids (UFA). The latter group was composed of monounsaturated (MUFA) and polyunsaturated fatty acids (PUFA). The content of the fatty acids representing the three mentioned classes differed significantly in the kernels of the analysed wheat cultivars (Table 2). The highest content of SFA was determined in kernels of cv. Muszelka, Bogatka, and Ostroga (Table 3). Small quantities of these compounds were determined in kernels of cv. Julius and KWS Livius. The content of unsaturated fatty acids (UFA) in wheat kernels of the tested cultivars followed a similar pattern. Their highest amounts were determined in kernels of cv. Bogatka and Ostroga, and these substances were the least abundant in kernels of cv. KWS Livius and Julius. Monounsaturated fatty acids (MUFA) appeared in large amounts in kernels of cv. Bogatka, Ostroga, and Bamberka. Less of these compounds were in kernels of cv. Julius. The highest quantities of PUFA were determined in kernels of cv. Bogatka, while low amounts of these fatty acids were noted in kernels of cv. KWS Livius and Julius.

The calculated value of the Pearson’s linear correlation coefficient *r* showed no significant correlations between the content of the fatty acids in wheat grain and the parameters describing the development of *S. granarius* populations (Table 4). Nevertheless, a significant positive correlation was demonstrated between the abundance of the pest’s offspring generation (*r* = 0.67, *p* = 0.05) and the total content of fatty acids in wheat grain. The content of individual compounds was positively correlated with the above parameter, hence a rise in the total content of fatty acids in wheat grain was correlated with an increase in the number of offspring specimens of the grain weevil. Statistical significance was only determined for one substance, which was C 20:1 (*r* = 0.72, *p* = 0.03). A similar type of correlation (lack of statistical significance) was noted for the mass of dust and loss of the mass of grain. In the latter case, a statistically significantly larger loss of grain mass was observed in connection with the content of C 17:1 (Table 4).

The analysed parameters describing the development of the grain weevil were also positively correlated with the growing content of compounds classified as saturate fatty acids (SFA) and unsaturated fatty acids (UFA: MUFA and PUFA). Ordinance techniques were also used to assess the relationship between the parameters describing the development of *S. granarius* and the content of the tested fatty acids (redundancy analysis—RDA). This type of analysis puts in order samples along a gradient, which is represented by an axis of an ordinance diagram, in this case, based on data on the content of fatty acids in wheat kernels and the parameters describing the development of *S. granarius*. Interpretation of this diagram implicated a strong correlation between the variables, such as the number of individuals in the pest’s offspring generation, the mass of dust, and loss of grain mass, versus the first ordinance axis (Figure 2).

This axis is also strongly correlated with the vectors representing the total content of fatty acids in kernels and the content of unsaturated fatty acids (UFA: MUFA and PUFA).

## 4. Discussion

Although the 21st century is called “the century of biologically active natural substances” [37], our knowledge about the influence of fatty acids contained in cereal kernels on the foraging by storage pests seems to be still insufficient. Information concerning these substances most often focuses on their role in shaping the basic physiological functions of the human body [3,38], and, increasingly often, of animals [39]. The understanding of mechanisms through which fatty acids act on insects remains fragmentary. The mode fatty acids from leaf surface waxes of *Ludwigia octovalvis* (Jacq.) P.H. Raven attract and stimulate insects has been examined by Mitra et al. [40]. These researchers proved that the most frequently present fatty acids were henicosanoic acid (C 21:0), palmitic acid (C 16:0), and docosanoic acid (C 22:0). Similar observations were reported by Adhikary et al. [41], who demonstrated the effect of volatile fatty acids present in seeds of *Lathyrus sativus* L. on the foraging by *Callosobruchus maculatus* (F.). Moreover, they observed the attraction of synthetic blends comparable to the composition of fatty acids in *L. sativus* of the four analysed cultivars by demonstrating the attractive effect of myristic acid (C 14:0), palmitic acid (C 16:0), and stearic acid (C 18:0) on *C. maculatus*. Krzyżowski et al. [42] demonstrated a significant effect of short-chain fatty acids on the physiology and behaviour of *C. maculatus* on stored leguminous plants. The biological assays they carried out showed that all the tested volatile fatty acids (especially propionic and valeric fatty acids) produced a significant toxic effect. González-Thuillier et al. [43] attest that C 16:0 and C 18:2, as well as their combinations, are the most abundant fatty acids in wheat grain. Similar conclusions can be drawn from the experiment reported in this paper, where it was shown that C 18:2, C 16:0, and C 18:1 were the most abundant fatty acids in wheat kernels (Table 3).

In the conducted study, no significant relationships were found between the content of fatty acids in the grain of the studied wheat cultivars and the parameters related to the development of *S. granaries*—the number of the progeny, the mass of dust produced, and the loss of grain mass (Pearson’s coefficient *r*). However, it was found that the total content of these substances in grain is positively correlated with the more numerous progeny of grain weevil. This may prove the influence of fatty acids contained in plants on the development of herbivores [40,41,42,43].

The current references concerning the effect of fatty acids in whole kernels of wheat on the behavioural and physiological parameters of the grain weevil provide rather fragmentary knowledge. The available information suggests that the content of lipids in most cereal plants plays a certain role in shaping the stability of storing cereal products and can influence the foraging by *S. granarius* [27,44] on stored cereal grain and products. Nawrot et al. [45] proved an effect of lipid fractions contained in kernels on the behaviour of *S. granarius*, *R. dominica*, and larvae of *Trogoderma granarium* (Everts). The data obtained from that experiment also showed that the content of fatty acids in wheat kernels had some influence on the activity of the grain weevil, and an increase in the total content of these compounds was closely correlated with a rise in the number of *S. granarius* offspring specimens. Liu [27] concluded that the content of fatty acids in grain from cultivars of the same species was negligible. Contrary conclusions were drawn by Narduccii et al. [31], who suggested that levels of the content of fatty acids in different wheat cultivars were different and depended on several factors. Similar observations were made during our experiment. The distribution of fatty acids was varied among the ten analysed wheat cultivars. The highest FA content was determined in cv. Bogatka (12.93%), Muszelka (12.44%), Ostroga (11.58%), and Bamberka 10.40%), while the lowest one was found in kernels of cv. Julius (5.96%) among ten tested wheat cultivars. The studied cultivars that had the highest fatty acid content in their kernels were the ones where the development of *S. granarius* was most intensive (Table 3 and Table 4). Evaluation of the dependence between the content of determined fatty acids and parameters describing the development of *S. granarius* was based on ordinance techniques (RDA). This analysis revealed a correlation between the parameters describing the development of *S. granarius* and the total content of fatty acids in kernels and the content of unsaturated fatty acids (UFA: MUFA and PUFA).

Moreover, the vectors illustrating the intensity of the grain weevil’s development were strongly correlated with a high content of compounds that belonged to the following groups: C 14:0, C 18:1, C 18:2, C 20:1, and C 22:0. In their study, Kordan et al. [17] observed an effect of fatty acids on basic life functions of the grain weevil, demonstrating, for example, that the extent of damage to the wheat kernel’s endosperm was positively correlated with the presence of fatty acid C 20:1, whereas the survivability of *S. granarius* was connected with the occurrence of fatty acids C 18:1 and C 18:2. Similar dependencies were observed during our experiment. The statistical analysis, values of Pearson’s correlation coefficient *r*, and RDA results suggest that the intensity of the grain weevil’s development is positively correlated with the increasing (10.40–12.93%) content of fatty acids (SFA and UFA) in wheat kernels. An important role, as substances stimulating the development of the pest insect, is probably played by compounds representing group C 20:1. Low content (5.96–7.55%) of fatty acids, in turn, is associated with a poorer development of these beetles. Such dependence, especially for cultivars Speedway, KWS Livius, and Julius, seems to be confirmed by the low content of C 15:0 (up to 0.0094%), C 16:1 (up to 0.0158%), and C 18:3 (up to 0.3275%) in grain produced by these cultivars.

## 5. Conclusions

The highest natural resistance to the feeding of *S. granarius* was found in five wheat cultivars: Julius, Platin, Speedway, Bogatka, and KWS Livius. The analysed kernels of winter wheat cultivars differed in the content of fatty acids, representing saturated (SFA) and unsaturated fatty acids (UFA). It was demonstrated that the intensity of the development of *S. granarius* is significantly correlated with the increasing content (above 2.0 mln) of saturated (SFA) and unsaturated fatty acids (UFA: MUFA and PUFA) in kernels. Compounds of the groups C 18:1 and C 20:1 probably play an important role as substances stimulating the development of this pest. The fatty acids that may act as deterrents to this species of pest insect are compounds C 15:0 (up to 0.01%), C 16:1 (up to 0.015%), and C 18:3 (up to 0.33%), which were determined in low amounts in kernels of the wheat cultivars Speedway, KWS Livius, and Julius. Considering the above results, cultivars with small amounts of fatty acids should be selected in the process of breeding new wheat cultivars.

## Figures and Tables

**Figure 1 insects-12-00806-f001:**
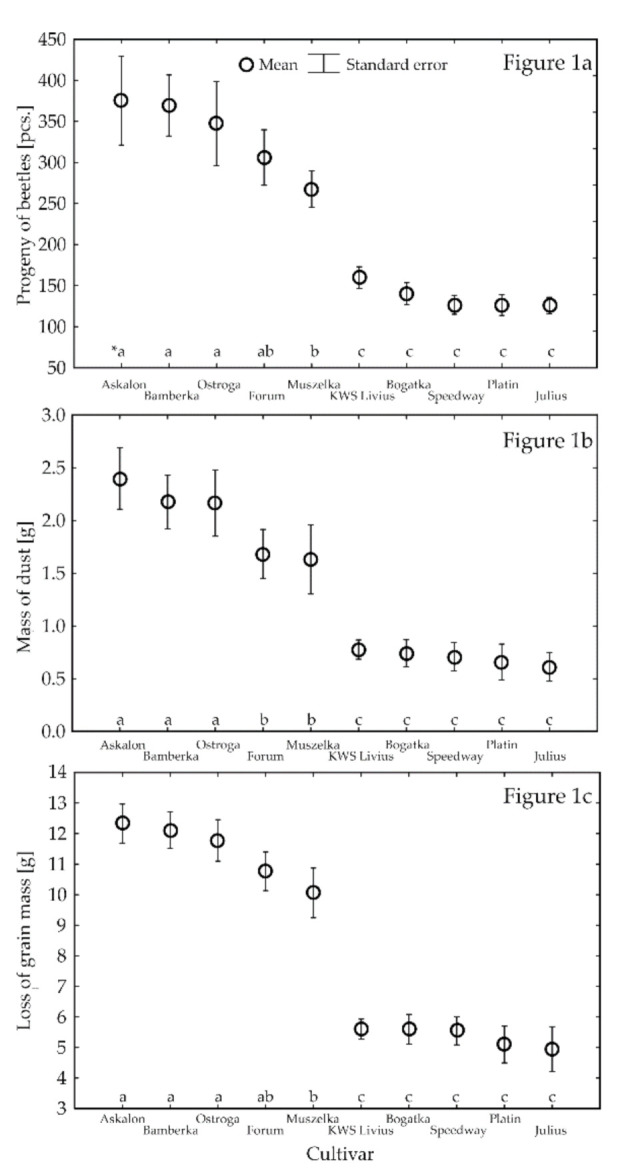
The average number of progeny beetles of *S. granarius* (**a**), the mass of produced dust (**b**), and the grain mass loss (**c**) observed on the grain of the studied wheat cultivars (* means followed by the same letter do not differ—Tukey’s HSD test).

**Figure 2 insects-12-00806-f002:**
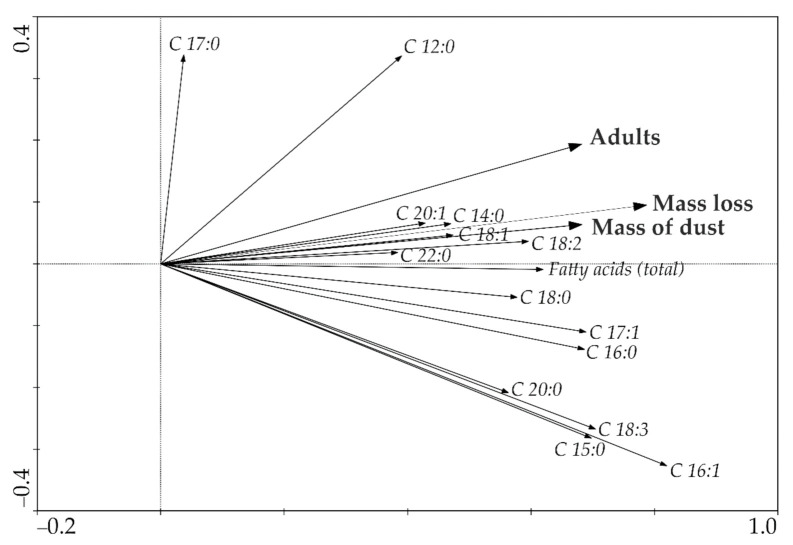
Redundancy analysis diagram (RDA) presenting correlations between analysed parameters pertaining to the development of *S. granarius* and the content of fatty acids in the analysed wheat cultivars.

**Table 1 insects-12-00806-t001:** Results of the generalised linear model (GLM) analysis for the number of *S. granarius* offspring specimens, the mass of produced dust and loss of the mass of kernels of the analysed wheat cultivars.

	df	Wald’s Statistic	*p*
Progeny of beetles	9	4227.4	0.00
Mass of dust	9	31.82	0.00
Loss of grain mass	9	108.53	0.00

**Table 2 insects-12-00806-t002:** Results of statistical assays (GLM, ANOVA) for the content of fatty acids in kernels of the analysed wheat cultivars.

Fatty Acids	df	Wald’s Statistic	ANOVA	*p*
			F Value	
C 12:0	9	4835.5	-	0.00
C 14:0	9	-	24.60	0.00
C 15:0	9	-	57.14	0.00
C 16:0	9	-	193.84	0.00
C 16:1	9	-	91.86	0.00
C 17:0	9	-	22.56	0.00
C 17:1	9	-	14.87	0.00
C 18:0	9	-	64.94	0.00
C 18:1	9	-	228.10	0.00
C 18:2	9	-	290.11	0.00
C 18:3	9	-	357.57	0.00
C 20:0	9	-	49.08	0.00
C 20:1	9	-	87.70	0.00
C 22:0	9	2835.0	-	0.00

**Table 3 insects-12-00806-t003:** The mean content and % of fatty acids in kernels of the analysed wheat cultivars.

Fatty Acids	Cultivar									
	Bogatka	Muszelka	Ostroga	Askalon	Bamberka	Forum	Speedway	Platin	KWS Livius	Julius
C 12:0	566.3 * bc **	551.3 bc	650.0 b	507.7 bc	1569.3 a	522.3 bc	485.0 bc	519.3 bc	415.0 bc	305.3 c
%	0.0029	0.0034	0.0029	0.0081	0.0026	0.0027	0.0027	0.0025	0.0022	0.0016
C 14:0	3496.3 a	2880.3 b	2793.3 bc	2320.3 cd	3024.7 ab	2149.3 d	2490.0 c	2578.7 c	1810.0 de	1703.0 e
%	0.0181	0.0145	0.0149	0.0157	0.0120	0.0134	0.0111	0.0129	0.0094	0.0088
C 15:0	2447.0 b	2950.3 a	2756.7 ab	2453.7 b	2522.3 b	1817.0 d	2281.7 c	2365.3 c	1802.7 d	1290.0 e
%	0.0127	0.0143	0.0153	0.0131	0.0127	0.0123	0.0094	0.0118	0.0093	0.0067
C 16:0	437,622.7 ab	458,877.7 a	448,480.0 a	379,471.0 c	416,400.3 b	329,163.7 d	371,442.0 c	384,799.3 c	282,869.0 e	227,370.7 f
%	2.2696	2.3259	2.3798	2.1595	1.9680	1.9956	1.7071	1.9264	1.4670	1.1792
C 16:1	3540.7 cd	5198.0 a	3921.3 bc	3952.7 b	3709.3 c	2933.7 f	3444.3 d	3421.0 d	3039.0 e	2263.3 g
%	0.0184	0.0203	0.0270	0.0192	0.0205	0.0177	0.0152	0.0179	0.0158	0.0117
C 17:0	4400.0 b	3728.3 d	3571.3 d	3569.3 d	4849.0 a	3911.3 cd	3998.0 c	4594.3 ab	4199.0 bc	2667.3 e
%	0.0228	0.0185	0.0193	0.0251	0.0185	0.0238	0.0203	0.0207	0.0218	0.0138
C 17:1	3316.3 b	4004.3 ab	4288.3 a	3276.0 b	3513.0 b	1747.3 d	2742.7 bc	1710.3 d	2349.7 c	1071.3 e
%	0.0172	0.0222	0.0208	0.0182	0.0170	0.0089	0.0091	0.0142	0.0122	0.0056
C 18:0	31,142.3 ab	28,317.7 b	31,775.7 a	25,101.3 bc	30,424.0 ab	22,978.0 d	30,079.7 ab	22,330.3 de	19,327.0 e	16,289.0 f
%	0.1615	0.1648	0.1469	0.1578	0.1302	0.1158	0.1192	0.1560	0.1002	0.0845
C 18:1	334,102.7 a	274,529.7 c	336,239.7 a	259,780.7 c	332,219.3 a	221,245.3 d	311,575.0 b	215,814.0 d	230,121.3 d	162,116.0 e
%	1.7327	1.7438	1.4238	1.7229	1.3473	1.1193	1.1474	1.6159	1.1935	0.8408
C 18:2	1,546,602.0 a	1,315,830.0 c	1,436,471.0 b	1,189,117.0 d	1,300,090.0 c	977,029.0 f	1,065,706.0 e	1,053,818.0 ef	830,340.0 g	670,930.0 h
%	8.0210	7.4498	6.8241	6.7425	6.1670	5.4653	5.0671	5.5269	4.3063	3.4796
C 18:3	98,222.7 c	111,380.0 a	100,923.7 bc	106,641.7 b	86,550.3 d	63,154.0 g	73,362.7 f	79,455.7 e	57,588.0 g	47,080.0 h
%	0.5094	0.5234	0.5776	0.4489	0.5531	0.4121	0.3275	0.3805	0.2987	0.2442
C 20:0	3915.3 a	3821.3 a	4137.7 a	4165.7 a	4010.7 a	3101.3 b	4229.7 a	3018.3 b	2769.3 b	2219.7 c
%	0.0203	0.0215	0.0198	0.0208	0.0216	0.0157	0.0161	0.0219	0.0144	0.0115
C 20:1	17,525.7 b	15,021.7 c	17,286.7 b	19,066.7 a	18,333.7 ab	12,165.0 d	14,939.3 c	12,325.7 d	15,115.3 c	9016.0 e
%	0.0909	0.0897	0.0779	0.0951	0.0989	0.0639	0.0631	0.0775	0.0784	0.0468
C 22:0	6259.3 ab	5814.3 ab	5912.7 ab	5944.3 ab	6747.3 a	5253.7 ab	5674.3 ab	6098.3 ab	4835.3 b	4031.0 c
%	0.0325	0.0307	0.0302	0.0350	0.0308	0.0316	0.0272	0.0294	0.0251	0.0209

* The area (arbitrary units) of the chromatographic peak representing the content of the fatty acid found. ** Means in rows followed by the same letter do not differ (Tukey’s HSD test).

**Table 4 insects-12-00806-t004:** Values of Pearson’s linear correlation coefficient *r* between the analysed parameters of the development of *S. granarius* population and the content of fatty acids.

	Adults		Mass of Dust		Mass Loss	
	*r*	*p* *	*r*	*p*	*r*	*p*
Adults	-	-				
Mass of dust	0.99	0.00	-	-		
Mass loss	0.98	0.00	0.99	0.00	-	-
C 12:0	0.55	0.1	0.51	0.14	0.53	0.12
C 14:0	0.19	0.60	0.18	0.63	0.26	0.48
C 15:0	0.45	0.19	0.49	0.15	0.55	0.1
C 16:0	0.41	0.23	0.43	0.21	0.50	0.14
C 16:1	0.44	0.2	0.49	0.16	0.57	0.09
C 17:0	0.3	0.93	−0.02	0.96	0.01	0.98
C 17:1	0.57	0.09	0.59	0.07	0.65	0.04
C 18:0	0.38	0.28	0.39	0.27	0.45	0.19
C 18:1	0.36	0.31	0.35	0.32	0.40	0.25
C 18:2	0.41	0.24	0.41	0.25	0.47	0.17
C 18:3	0.64	0.06	0.56	0.1	0.61	0.06
C 20:0	0.60	0.09	0.53	0.11	0.56	0.09
C 20:1	0.72	0.03	0.58	0.08	0.58	0.08
C 22:0	0.56	0.12	0.43	0.22	0.46	0.18
Fatty acids (total)	0.67	0.05	0.43	0.22	0.49	0.15

* The value of the test probability *p.*

## Data Availability

The development data for *S. granarius* presented in this study are available on request from the corresponding author.

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
