# Peer review of "Fatty Acids Present in Wheat Kernels Influence the Development of the Grain Weevil (Sitophilus granarius L.)"

_insects, 2021, doi:10.3390/insects12090806_

Round 1
Reviewer 1 Report
The topic is interesting with potential practical implications for plant breeding towards resistance to Sitophilus granarius, and the manuscript is generally well-written.
I think that the authors need to be careful about their use of the words ‘development’ and ‘development rate’, because the meanings used in this manuscript do not agree with convention. Most entomologists would understand development to be the progress through developmental stages, and rate of development would be understood as how quickly this occurred (normally 1/days or 1/weeks etc). The authors have recorded the number of offspring but not how quickly they developed, so the manuscript must be revised to remove the words ‘development’ and ‘rate of development’, and describe the study more accurately.
As well as the number of offspring, the authors also assessed damage to the wheat (as weight loss), fatty acid content of the various wheat varieties, and the relationship between fatty acids and these variables. A measure of development time on each wheat variety would have been valuable, as it may have varied across the varieties, so some justification for why this was not assessed is needed.
Lines 55-56: Although S. granarius is a serious pest, it is not found in all grain-growing regions. Therefore, this statement should be revised to explain which types of places (countries or climatic zones) it is most frequently found.
Line 73: Add ‘Sitophilus zeamais (Motschulsky)’.
Line 93: Delete ‘rate’.
Lines 100-101:
Lines 102-103:
Line 114: What temperature and humidity?
Lines 144-145: Provide some information on this population. Was it a long-standing laboratory strain? If not, provide some details about its collection (year, location).
Line 150: Use ‘Twenty’ at the beginning of a sentence.
Lines 150-151: Approximately how old were the adults e.g. days/weeks after eclosion?
Line 153: Why was 8 weeks chosen? It’s possible that egg-adult emergence time could vary between cultivars.
Line 195: Delete ‘specimens’
Lines 195-196: ‘three cultivars’
Correlations: The results of correlation analysis must be moved from the discussion section to the results section.
Discussion: The name (or recognised abbreviation) of the taxonomist is needed in several places e.g. in Line 257.
Reviewer 2 Report
The manuscript is well-conceived and well-written. There are a couple of minor changes that would improve the manuscript. The units in Table are not provided, they appear to be flame ionization counts. It would be much preferable to give the amounts of the fatty acids in units such as milligrams or as percentage of total fatty acids present. Also, the authors should provide a recommendation of which cultivars might be more resistant to attack (i.e., the five varieties with lowest number of progeny, grain dust, and grain mass lost). The conclusions should give some statement about how this data can be used in a real world situation.Author Response
Please check the attachment.

Reviewer 3 Report
Please read the comments accordingly carry out changes. In your experiments you do not mention any control using the normal wheat, why? Please explain.
